# Recurrent Neural Network to Predict Saccade Offset Time Points from Electrooculogram Signals for Automatic Measurement of Eye-Fixation-Related Potential

Takuma Saga [1,2,3], Hiroki Watanabe [1] and Yasushi Naruse [1,*]

1    Center for Information and Neural Networks, Advanced ICT Research Institute, National Institute of Information and Communications Technology and Osaka University, 588-2, Iwaoka, Iwaoka-cho, Nishi-ku, Kobe, Hyogo 651-2492, Japan
2    Graduate School of Frontier Biosciences, Osaka University, 1-3 Yamadaoka, Suita, Osaka 565-0871, Japan
3    Research Fellow for Young Scientists of the Japan Society for the Promotion of Science, 5-3-1, Kojimachi, Chiyoda-ku, Tokyo 102-0083, Japan
*    Correspondence: y_naruse@nict.go.jp; Tel.: +81-78-969-2225

**Abstract:** Eye-fixation-related potential (EFRP)—an event-related potential that is time-locked to the saccade offset (SO)—can be measured without synchronizing with time when external stimuli occur. Such an advantage in measurement enables the mean amplitude of the EFRP to be used to estimate the cognitive workload, which is known to change the amplitude, under real-world conditions. However, to observe EFRPs reliably, the SO timing must be correctly and consistently determined in milliseconds owing to the high temporal resolution of the electroencephalogram (EEG). As the electrooculogram (EOG) is commonly measured simultaneously with the EEG and the SO timing is reflected as a steep change in the waveforms, attempts have been made to determine the SO timing from EOG signals visually (the VD method). However, the SO timing detected by the VD method may be inconsistent across trials. We propose a gated recurrent unit—a recurrent neural network model—to detect the SO timing from EOGs consistently and automatically. We used EOG data from a task that mimics visual inspections, in which participants periodically traversed their eyes from left to right, for the model training. As a result, the amplitudes of the EFRPs based on the proposed method were significantly larger than those based on the VD method and the previous automatic method. This suggests that the proposed method can prevent the decrease in EFRP amplitudes owing to the inconsistent determination of the SO timing and increase the applicability of cognitive workload estimation using the EFRP in real-world environments.

**Keywords:** electrooculogram; eye-fixation-related potential; gated recurrent unit; passive brain–computer interface; saccade offset

## 1. Introduction

Measurements of electroencephalogram (EEG) signals have been applied to estimate human mental states such as the depressed state [1] and motivation [2]. In these studies, event-related potentials (ERPs) were used for estimating the human mental state. ERPs are EEG responses that are time-locked to the onsets of the events such as visual and/or auditory stimuli. As the amplitudes of ERPs are commonly small compared to the noise and spontaneous activity, many EEG epochs that are time-locked to the onset of events must be averaged to detect the ERPs. Therefore, it is necessary to know the timing of the events to use the ERPs for estimating the mental state. However, it is difficult to determine exactly when people watch or listen in the real environment. Thus, the use of ERPs for mental state estimation in real environments has been limited. However, Wunderlich et al. succeeded in measuring the eye-fixation-related potential (EFRP), which is a type of ERP, during walking in a real environment [3].

The EFRP is observed synchronously with saccade offset (SO) timing [4]. As the SO timing coincides with the onset time of perceiving a visual object, the timing can be used as the onset of the timing when the objects are perceived, even in real environments. In particular, the lambda response, which is part of the EFRP, is a positive component that is observed in the occipital region approximately 80 ms after the SO timing [4]. The amplitude of the lambda response is known to change according to the workload [5] and attention levels that are dedicated to visual objects [6]. Owing to these characteristics, the lambda response can be used as an indicator to predict the occurrence of visual inspection errors that are caused by changes in visual attention to the inspected object as a result of prolonged inspection [7]. Furthermore, it has been applied to workload estimation while driving a car [8].

Electrooculograms (EOGs) or eye-tracking data have commonly been used to detect SO timing [3–5]. It is advantageous to use EOG data to determine the SO timing for measuring the EFRP because the same recording systems can measure both EEG and EOG data. Moreover, the EOG can be measured by simply placing electrodes around the eyes. In previous research using EOG data for this purpose, the SO timing was determined visually as the time point of change from saccade to fixation [9]. However, this visual determination (VD) method is time-consuming and restricted to offline analysis. Therefore, the SO timing must be determined automatically to use the EFRP responses to estimate mental states in real time. Moreover, the SO timing may be inconsistent among detectors partially due to fatigue owing to long periods of visual inspection, especially when analyzing EFRPs with large datasets. The inconsistency of the SO timing determination causes decreases in the EFRP amplitude in the averaged EEG data over EEG epochs that are time-locked to the determined SO timing [10]. Therefore, the improvement in consistency by automating the determination of the SO timing will expand the applicability of state estimation in real-world settings using EFRP.

A previous study proposed an automatic method that can detect saccade timing to obtain EFRP responses using the EOG signal [3]. In this method, independent components (ICs) reflecting vertical and horizontal EOGs are first identified using independent component analysis (ICA). Next, the SO timings are detected based on the peak velocity of these ICs (hereafter, this previous method is referred to as the peak velocity (PV) method). Using this method, the authors successfully observed EFRPs from EEG data that were measured in the real environment, i.e., while walking in the city.

Although the PV method enables the saccade timing to be detected automatically in real environments, applying neural network (NN) models to detect SO timing may improve performance. In recent years, NN models have been used to analyze EEG and EOG data for various purposes. For example, the use of recurrent neural network (RNN) models for EEG and EOG data improved the accuracy of detecting drowsiness while driving [11]. Furthermore, a combination of convolutional neural networks (CNNs) and RNN models was used for EOG data to improve the accuracy of sleep stage classification [12]. Thus, NN models have been applied to EEG and EOG data and have improved the classification accuracies compared to traditional methods. Among the NN models, RNN models have also demonstrated potential for change point detection. For example, an RNN model was proposed to predict the change points from data of sensors attached to industrial equipment parts [13]. Moreover, an RNN model outperformed the previous method in detecting the change points in multiple time-series load data of an electric power company for power outage analysis [14]. As the SO timing can be considered as a change point from saccade to fixation detection, these previous studies have suggested that RNN models can also be applied to SO timing detection. In general, a large amount of training data are required to improve the detection accuracy of NN models. As a large amount of EOG and EEG data are rarely available for SO timing detection, to the best of our knowledge, no study has applied an NN model to detect the SO timing automatically by relying only on EOG signals. In our previous study, we obtained a relatively large EEG and EOG dataset while participants periodically traversed their eyes from left to right in a task that mimicked visual inspection

(2500 trials with 50 participants in total) [7]. These relatively large datasets provide the possibility of using NN models to detect SO timing, which may yield better performance than the VD and PV methods.

The purpose of this study was to develop a novel NN model to predict the SO timing using the dataset that was obtained in the previous research [7]. The labels indicating the SO timing were created by the VD method and the labeled data were used for supervised learning. We employed a gated recurrent unit (GRU)—an RNN derivative model that is suitable for handling time series data [15]. In recent years, other models that are suitable for handling time-series data, such as the Transformer [16] and Seq2Seq [17], have been proposed. However, these new methods are more complex than RNN models [16,17]. Seq2Seq uses RNNs as encoders and decoders, and, therefore, it is computationally heavier than RNN models. The heavy computational cost is also an issue in the Transformer because it includes tens of millions of parameters [18]. As this is the first study to detect saccade offsets automatically using NN models, we used an RNN model, which is relatively computationally light and easy to apply.

The trained RNN model was evaluated in terms of the (1) error between the SO timing determined by the proposed and VD methods and (2) the consistency of the detected SO timing across trials. For the first evaluation procedure, we verified whether the proposed RNN model correctly predicted the SO timing by calculating the difference between the SO timing that was predicted by the proposed model and that obtained by the VD method. The second evaluation procedure was intended to account for the possible inconsistency in the SO timing across trials, which may occur owing to fatigue or spreading the work over several days because of the lengthy determination process. As ERPs are observed by averaging multiple trials relative to the event onset timing, the consistent determination of the SO timing increases the mean amplitudes. Conversely, inconsistent determination across trials decreases the mean amplitudes, which may affect the reliability of the use of EFRPs for mental state estimation. Thus, the evaluation of the inconsistency is important in terms of practical use. We adopted the mean amplitudes of the lambda response as the consistency index and compared the values across the proposed, VD, and PV methods.

## 2. Materials and Methods

### 2.1. Participants

In this study, we used the EOG and EEG data that were obtained in our previous study [7]. Data were collected from 50 participants (25 males and 25 females, age range: 20 to 39).

### 2.2. Data Measurement Equipment

EEG data from the FCz, Pz, O1, and O2 positions were measured based on the international 10-10 system using a wireless portable EEG device (PolymateMini AP108, Miyuki Giken Co., Ltd., Tokyo, Japan) and dry electrodes (Unique Medical Co., Ltd., Tokyo, Japan). The horizontal and vertical EOGs were measured from electrodes that were placed next to the lateral canthus of and above the eyebrow of the participant's left eye. The ground and reference electrodes were placed on the left and right mastoid, respectively. The sampling frequency of all signals was 500 Hz.

### 2.3. Task

In the previous study, participants performed a task that mimicked the visual inspection of printed circuit boards. In the task, monochrome and color images of printed circuit boards were displayed on the left and right sides of a monitor, respectively. The participants moved their eyes from left to right at predetermined fixed intervals to determine whether the right image differed from the left one. The visual angle between the two images was approximately 29°. The participants performed two sessions per day (625 trials/session) over two days (total = 2500 trials; for details, see [7]).

### 2.4. Data Analysis

#### 2.4.1. Preprocessing of EOG Data

The EOG data were preprocessed using Matlab version R2018a (MathWorks, Inc., Natick, MA, USA). A median filter with a width of 500 ms was applied to the EOG data to remove high-frequency noise [19]. The width of the median filter was determined visually to minimize fluctuations in the value of the static state of the EOG data. The first author determined the SO timing using the VD method with every saccadic shift from an object on the left board to that on the right. Trials in which the SO timing could not be determined owing to extreme noise caused by the equipment were excluded from the analysis. As some trials in which the SO timing could not be determined were excluded, the average number of trials that was used for the analysis was 1436.2 (standard deviation (SD) = 562.1) across all participants, and the total number of trials with all participants was 71,809. Because the EOG data used in the PV method were sampled at 250 Hz, the EOG data were downsampled to 250 Hz to compare the proposed method with the PV method [3].

The preprocessed EOG data were segmented into lengths of 400 ms (i.e., 101 time points), and one segmented data item was considered as a single trial. All trials included the SO timing determined by the VD method. However, the position of the SO within the trial was randomly varied per trial. This procedure was applied because, if the position of the SO timing in the trials was biased (e.g., the SO was concentrated around a center time point within trials), the proposed model would learn the bias rather than the point when the saccade changed to fixation. The label data for the trial were one-hot vectors in which the point of the SO timing was 1 and the other points were 0.

#### 2.4.2. Proposed Method

The structure of the proposed GRU model is depicted in Figure 1a. The GRU consists of a reset gate that determines the amount of past information to be discarded, an update gate that determines the amount of past information to be passed to the future, candidate output, and output. The reset gate ($r^{(t)}$), candidate output ($h^{(t)}$), update gate ($z^{(t)}$), and output ($y^{(t)}$) are defined in Equations (1)–(4), respectively.

$$r^{(t)} = f_r^{\sigma}\left(x^{(t)}W_r + y^{(t-1)}V_r + b_r\right), \tag{1}$$

$$h^{(t)} = f_h^{tanh}\left(x^{(t)}W_h + (r^{(t)} \odot y^{(t-1)})V_h + b_h\right), \tag{2}$$

$$z^{(t)} = f_z^{\sigma}\left(x^{(t)}W_z + y^{(t-1)}V_z + b_z\right), \tag{3}$$

$$y^{(t)} = \left(1 - z^{(t)}\right) \odot y^{(t-1)} + z^{(t)} \odot h^{(t)}, \tag{4}$$

where $x^{(t)}$ is the input vector at time t, $W$ is the weight matrix of $x^{(t)}$ for each gate, $V$ is the weight matrix of $y^{(t)}$ for each gate, and $b$ is the bias vector input to each gate. Furthermore, $f^{\sigma}$ is the sigmoid function and $f^{tanh}$ is the hyperbolic tangent function. The operation symbol $\odot$ indicates the Hadamard product.

The architecture of the proposed GRU model is illustrated in Figure 1b, where $T$ represents the number of time points and $C$ denotes the number of input data dimensions. As the input data were single-trial EOG data, $T$ was 101. Furthermore, C was 2 because the EOG data consisted of two channels. The $H$ in the hidden layer and 1 in the output layer are the number of neurons in each GRU. The hyperbolic tangent function was used as the activation function for the output of GRU 1, and the softmax function was used as the activation function for the output of GRU 2. The model was trained using Adam, which is a stochastic gradient descent algorithm, to minimize the cross-entropy error between the output of Layer 2 and the label data of the input data (epoch number: 25, batch size: 256). The optimal number of $H$, which is a hyperparameter, was estimated using a grid search

from a range of 25 to 150 (in increments of 25) based on the mean absolute error between the label data and the output of Layer 2 in three-fold cross-validation using the training data. The model was evaluated using leave-one-subject-out cross-validation.

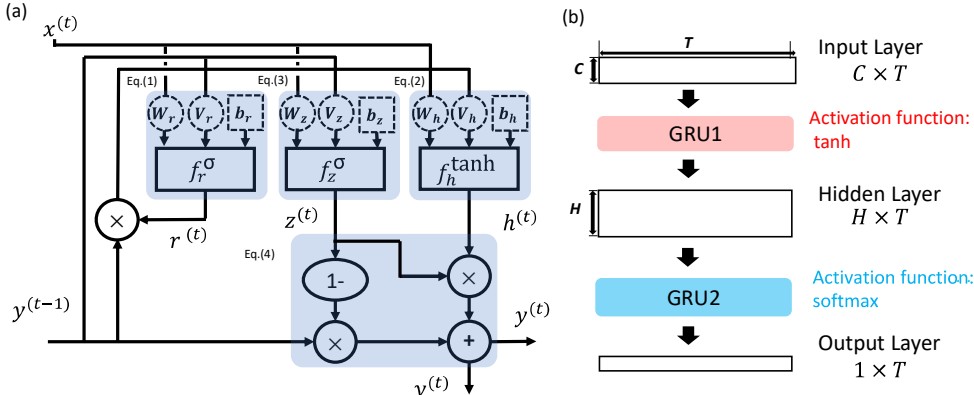

**Figure 1.** Description of the proposed model. (**a**) Structure of GRUs corresponding to Equations (1)–(4). The corresponding parts of Equations (1)–(4) are highlighted in blue. (**b**) Network structure of the proposed model.

### 2.4.3. Implementation of PV Method

To the best of our knowledge, the PV method is the only method that detects the saccade timing from the EOG signal only and observes the EFRP. We implemented the PV method that was proposed in [3]. All parameters in this section have the same values as in the previous study. First, ICA was applied to all measured data, and after visually extracting the IC data corresponding to eye movements, the following equation was applied:

$$EOG = \sqrt{IC_{EOG_v}^2 + IC_{EOG_h}^2}, \tag{5}$$

where $IC_{EOG_v}$ indicates the IC data corresponding to the vertical eye movement and $IC_{EOG_h}$ indicates the IC data corresponding to the horizontal eye movement. Subsequently, the square of the EOG velocity was calculated by taking the derivative, as follows:

$$dEOG = diff(EOG)^2, \tag{6}$$

where $diff$ is the derivative. A median filter with a width of 80 ms was applied to the $dEOG$, and the peak point was defined as the saccade time. Note that the PV method detects the saccade time, which is the timing of the peak velocity, rather than the SO timing.

If the saccade was detected within $\pm120$ ms relative to the blink timing, the detected saccade point was excluded, as in [3]. The median filter with a width of 80 ms was applied to the $IC_{EOG_v}$ data, following which, the peak point was defined as the blink time, to detect the blink.

As we only focused on left-to-right eye movements, we used only the $IC_{EOG_h}$ data for the saccade time detection. The *findpeaks*() function (see https://www.mathworks.com/help/signal/ref/findpeaks.html (accessed on 1 April 2023) for details), which is a built-in function in Matlab version R2018a that is designed to determine local maxima, was used for the peak detection. Table 1 presents the parameters of the *findpeaks*() function for the saccade and blink time detection in the PV method.

**Table 1.** Summary of parameters of *findpeaks*() functions used in the PV method.

|  | MinPeakWidth | MaxPeakWidth | MinPeakDistance | MinPeakHeight | MinPeakProminence |
|---|---|---|---|---|---|
| Blink time | 20 ms | 320 ms | 100 ms | 90-percentile of *dEOG* | 85-percentile of *dEOG* |
| Saccade time | 4 ms | 40 ms | 100 ms | 90-percentile of *dEOG* | 90-percentile of *dEOG* |

To compare the performance of the proposed model directly with that of the VD and PV methods, their performances were evaluated using the same trials. Thus, among the saccades that were detected by the PV method, the saccades within ±100 ms relative to the SO timing determined by the VD method were considered to correspond to the same saccade across the two methods, and we used trials including the saccades for the following analysis. The time interval was determined considering the time lag between the saccade time detected by the PV method and the SO timing determined by the VD method, as the PV method detects the time point when the velocity of the saccade is maximal, rather than the SO timing.

2.4.4. Calculation of Mean Amplitude of Lambda Response

A band-pass 3000-order finite impulse response filter of 1 to 15 Hz was applied to the EEG data [20]. The filtered EEG data were extracted in the range of $-600$ to $600$ ms based on the SO timings determined by the VD and proposed methods and the saccade time detected by the PV method. We calculated the average waveform over the extracted EEG data measured from the O1 and O2 electrodes. Baseline correction was performed using a mean amplitude from $-600$ to $-500$ ms. Extracted EEG data that included any time points exceeding ±80 μv were removed. Thus, a total of 54,177 extracted EEG data were used for the following analyses.

The mean amplitude of the lambda response was calculated for each participant using a 24 ms time window that was centered on a peak latency of the lambda response of the grand-average waveform. The peak latency of the grand-average waveform was 84 ms.

*2.5. Statistical Analysis*

We calculated the differences between the SO timing detected by the proposed and VD methods to evaluate whether the proposed method learned the SO timing determined by the VD method. If the proposed method could not learn the SO timing determined by the VD method, the difference would result in a uniform distribution. Thus, we performed the Kolmogorov–Smirnov test to determine whether the values differed significantly from the uniform distribution.

Thereafter, paired *t*-tests were performed to compare the mean amplitude of the lambda response based on each of the three methods to evaluate the consistency of the detected SO timing. The significance level was 0.05. The *p*-values were corrected using the Bonferroni method for multiple comparisons.

**3. Results**

First, we examined whether the proposed model learned the SO timing determined by the VD method. The mean difference was 5.07 ms (SD = 24.0). In 98.6% of the trials (53,432 trials; all trials used for the analysis = 54,177), the difference in the SO timing between the proposed and VD methods was within ±50 ms. Figure 2 displays a histogram of the difference for all trials. The Kolmogorov–Smirnov test demonstrated that the distribution of the difference deviated significantly from the uniform distribution (D = 0.543, $p < 0.001$). This result indicates that the proposed model successfully learned the SO timing of the VD method.

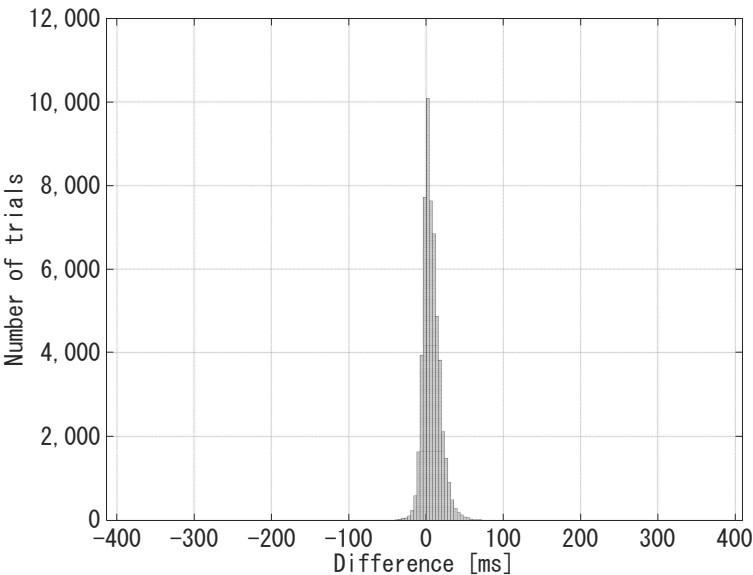

**Figure 2.** Histogram of the difference between SO timing detected by the proposed and VD methods.

Figure 3 shows the grand-average EFRP waveforms based on the three methods. Note that each waveform was shifted to align the peaks of the lambda response, as the PV method detects the saccade time rather than the SO timing. The lambda response was identified at approximately 80 ms in all methods. Figure 4 shows the mean amplitude of the lambda response based on each method. The mean amplitude was 0.050 μV (SD = 1.10) for the PV method, 0.273 μV (SD = 1.30) for the VD method, and 0.545 μV (SD = 1.26) for the proposed method. The mean amplitude of the lambda response based on the proposed method was significantly larger than that based on the VD method (t(49) = 2.88, $p$ = 0.018 (Bonferroni-corrected)) and that based on the PV method (t(49) = 5.03, $p$ = 2.12 × $10^{-5}$ (Bonferroni-corrected)). The larger mean amplitude of the lambda responses in the proposed method suggests that the SO timing detected by the method was more consistent across trials compared to those detected by the VD and PV methods. There was no significant difference between the VD and PV methods (t(49) = 1.77, $p$ = 0.250 (Bonferroni-corrected)).

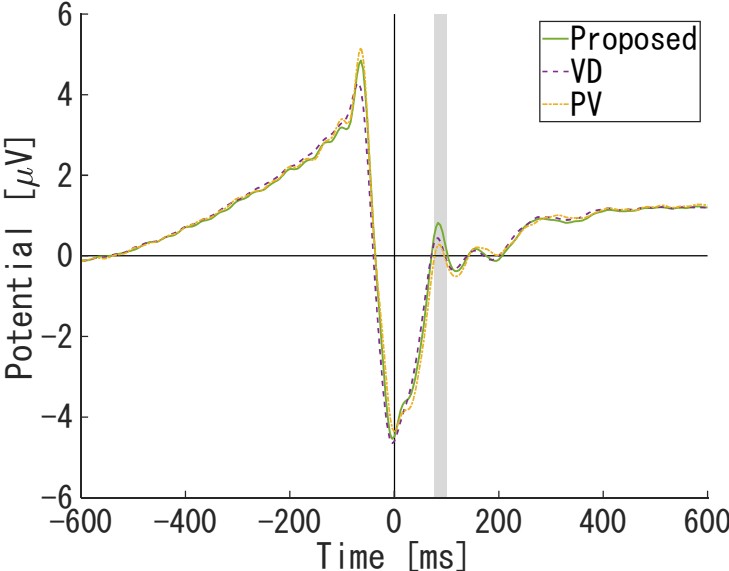

**Figure 3.** Grand-average waveforms based on the three methods. Each waveform was shifted to align the peaks of the lambda response.

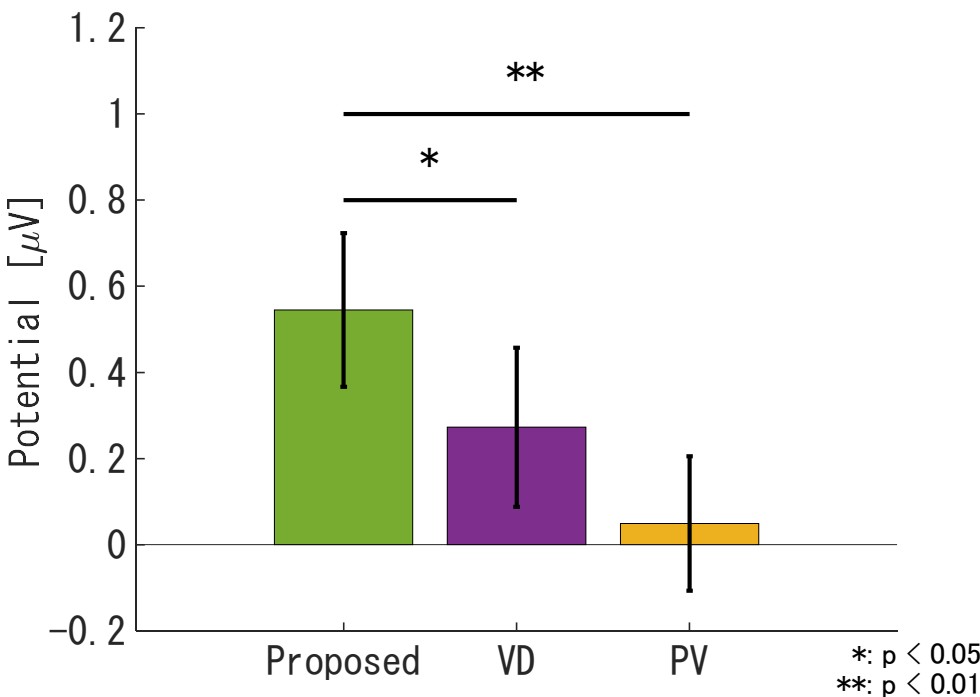

**Figure 4.** Comparison of the mean amplitude of the lambda response across the three methods. The error bars indicate the standard errors.

## 4. Discussion

In this study, we proposed a novel RNN-based model to detect the SO timing automatically from EOG signals alone. The histogram of the differences between the predicted SO timing using the proposed method and that determined using the VD method indicated that the proposed model learned the SO timing determined by the VD method. This result, together with the observation of clear EFRP responses with the SO timing predicted by the proposed method, demonstrates that the proposed model can automatically detect the SO timing for observing EFRPs. The detection process needs to be automated, especially when using a large dataset (2500 trials per person × 50 participants = 125,000 trials in total), as in this study. Furthermore, automated detection allows the response to be measured online in a real-world environment. Thus, our proposed automated determination of EFRPs will expand the availability of EFRP responses for mental state estimation in real environments.

The mean amplitude of the lambda response based on the proposed method was significantly larger than that based on either the PV or VD method. As the lambda response was obtained by averaging across trials that were time-locked to the SO timing, an increase in the consistency of the SO timing prediction across trials theoretically increases the amplitude of the lambda response in the EFRP waveform. That is, as the consistency of the SO timing decreases, the mean amplitude decreases [10]. Such inconsistent decisions probably occur owing to the fatigue caused by many hours of determination by visual inspection. The enhanced mean amplitudes of the lambda responses based on the proposed method indicate that it provides more consistent prediction results across trials than the VD method, which is susceptible to operator fatigue during the decision process, and thus, automatic detection by our proposed model enables the EFRP amplitudes to be calculated more reliably.

The most consistent prediction may have benefited from the more complex model architecture of the proposed RNN-based method compared to the PV method. Conversely, as a positive correlation generally exists between the model complexity and the required volume of training data, a very large quantity of data is required for training NN models because of the many model parameters that are required to realize the model complexity [21]. This study used a biometric dataset of 54,177 trials, which is relatively large. This large data

volume enabled the training of the proposed model, and the more complex model architecture compared to the PV methods may have contributed to more consistent prediction. Moreover, as noted previously, the PV method detects the saccade time based on the peak velocity rather than the SO timing. However, EFRP responses are observed time-locked to the SO timing and not the saccade time. The proposed method offers the advantage of detecting the SO timing, which may allow for more consistent prediction compared to the PV method.

It may seem paradoxical that the proposed model that was trained by the SO timing determined by the VD method improved the consistency of the prediction compared to the VD method. However, as mentioned previously, the VD method is likely to be inconsistent in determining the SO timing owing to fatigue. These inconsistent label data can be considered as noise when training the proposed model. We believe that the proposed model is sufficiently generalizable to perform robust predictions against such noise by using a large amount of data including noise, thereby outperforming the VD method. In general, various types of noise tend to contaminate biological data, especially when they are recorded in a real environment [3]. The results of this study suggest that learning the nature of biological data can be facilitated by training the NN model with large quantities of data.

The proposed method is limited by the fact that it only targets EOG data in which participants move their gaze from left to right. It is necessary to identify the SO timing from EOG data when participants move their gaze in any direction to apply the proposed method to the observation of EFRP responses in real environments. In the future study, we plan to obtain data on saccade in any direction and improve our model to detect the SO timing for any direction.

### 5. Conclusions

We have proposed an RNN-based model as the first NN model to detect the SO timing using only EOG signals. The proposed model learns the SO timing determined by the VD method and can automatically detect the SO timing to observe the EFRP. Furthermore, its performance is superior to that of the two conventional methods, and this result extends the applicability of using EFRPs for estimating mental states online in a real environment. We aim to improve the model for SO timing detection in various directions in the future.

**Author Contributions:** Conceptualization, Y.N.; methodology, Y.N. and T.S.; software, T.S.; validation, H.W.; formal analysis, T.S.; investigation, T.S.; resources, Y.N.; data curation, T.S.; writing—original draft preparation, T.S.; writing—review and editing, H.W. and Y.N.; visualization, T.S. and H.W.; supervision, Y.N.; project administration, Y.N.; funding acquisition, T.S., H.W. and Y.N. All authors have read and agreed to the published version of the manuscript.

**Funding:** This work was partially supported by JSPS KAKENHI, grant numbers JP20J20682, JP21H03573, and JP20K14110.

**Institutional Review Board Statement:** The study was conducted in accordance with the Declaration of Helsinki and approved by the National Institute of Information and Communications Technology (28 August 2018).

**Informed Consent Statement:** Informed consent was obtained from all subjects involved in the study.

**Data Availability Statement:** The data is unavailable due to ethical restrictions.

**Conflicts of Interest:** The authors declare no conflict of interest.

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
