# Peer review of "Recurrent Neural Network to Predict Saccade Offset Time Points from Electrooculogram Signals for Automatic Measurement of Eye-Fixation-Related Potential"

_applsci, doi:10.3390/app13106230_

Round 1

Reviewer 1 Report

This work describes the development of a neural network model to predict saccade offset time points. The study is interesting and the methodology followed seems appropriate, however, the following issues must be addressed before being considered for publication:

1) The caption in Table 1 must be replaced.

2) Group figures and use letters to refer to them.

3) The discussion and the conclusions should be separated and expanded.

Author Response

Responses to Reviewer 1 Comments

Point 1: Section 2.4.3: The caption in Table 1 must be replaced.

Response 1: We have corrected the caption of Table 1, as indicated below and highlighted in green in the text.

“Summary of parameters of findpeaks() functions used in the PV method.” (page 6, line 244)

Point 2: Section 2.4.2, 3: Group figures and use letters to refer to them.

Response 2: We have grouped Figures 1 and 2 into a new Figure 1. We have also added the below text and highlighted it in green.
“The corresponding parts of Equations (1) to (4) are highlighted in blue.” (page 5, line 207)

Point 3: Section 4: The discussion and the conclusions should be separated and expanded.

Response 3: We have separated the Discussion and Conclusions sections and have added contents. The revisions are highlighted in green. (pages 9, line 355)

Reviewer 2 Report

The paper " Recurrent neural network to predict saccade offset time points from electrooculogram signals for automatic measurement of eye-fixation-related potentials" is dedicated to comparing methods for detecting time points of saccade offset (SO). Authors proposed a model of recurrent neural network to determine SO time points from EOGs consistently and automatically. The main result of this work is that their approach is more accurate for determining SO time points than the two conventional methods.

I think that this approach to the definition of SO time points has a right to exist and can be useful for the analysis of event-related potential.

However, I have several comments:

·      First of all, I want to indicate that this paper is carelessly made. This can be seen from a following: inaccuracy (for example: «Data were collected from 50 participants (25 males, age range: 20 – 39)».  Who are the others 25 from the 50 participants?  One more:   «Table 1. This is a table. Tables should be placed in the main text near to the first time they are cited» ); Lack of a clear description of the used methods;  Poorly executed drawings (In particular, In 4 and 5 figures you can see pixels).

·      Authors used Matlab and findpeaks() function for processing EOG data. I believe that when writing scientific papers, you should not refer only to the name of the function without a reference to the methods it implements.   

·      It is not clear why the authors downsampled EOG data. However, it seems that the accuracy of timing plays a key role in the analysis of evoked potentials.

·      It is not clear from the paper on the basis of what considerations the parameters presented in Table 1 were chosen.

·      Based on Figure 5, the authors compared methods using statistical methods but they are not described anywhere in the paper. It is also unclear what the asterisks in this figure mean. I assume that the asterisks indicate the level of statistical significance but the paper should describe it.

I think the paper can be accepted after revision and considering all comments above.

Author Response

Responses to Reviewer 2 Comments

Point 1: Section 2.1: First of all, I want to indicate that this paper is carelessly made. This can be seen from a following: inaccuracy (for example: «Data were collected from 50 participants (25 males, age range: 20 – 39)». Who are the others 25 from the 50 participants?, Section 2.4.3: One more: «Table 1. This is a table. Tables should be placed in the main text near to the first time they are cited» ); Lack of a clear description of the used methods; Poorly executed drawings (In particular, In 4 and 5 figures you can see pixels).

Response 1: ・We have included the correct participant information. The revision is highlighted in green. (page 3, line 135)

・We have corrected the caption of Table 1, as indicated below, and highlighted it in green.

“Summary of parameters of findpeaks() functions used in the PV method.” (page 6, line 244)

・We have included information on the data preprocessing and PV methods. The corrected parts are highlighted in green in the text. (page 4, line 161; page 4, line 165; page 5, line 212; page 5, line 230)

・We have changed the figures from raster graphics to vector graphics. Although dots may appear depending on the display of Microsoft Word, these would disappear after conversion to PDF. (Figures 1 to 4)

Point 2: Section 2.4.3: Authors used Matlab and findpeaks() function for processing EOG data. I believe that when writing scientific papers, you should not refer only to the name of the function without a reference to the methods it implements.

Response 2: An explanation of the Matlab command is provided below, but we could not find a description of the algorithm to search for peaks or references. Thus, we have corrected the sentence by adding a summary and have highlighted it in green. (page 5, line 230)
―――――――――――――――――――――――――――― 
pks = findpeaks(data) returns a vector with the local maxima (peaks) of the input signal vector data. A local peak is a data sample that is either larger than its two neighboring samples or is equal to Inf. The peaks are output in order of occurrence. Non-Inf signal endpoints are excluded. If a peak is flat, the function returns only the point with the lowest index.

Point 3: Section 2.4.1: It is not clear why the authors downsampled EOG data. However, it seems that the accuracy of timing plays a key role in the analysis of evoked potentials.

Response 3: We agree with the comment that downsampling reduces the timing accuracy. Hence, in order to compare the accuracy with previous studies, downsampling was performed to obtain the same sampling rate as that in the previous studies. The reason for downsampling the EOG data has been included and highlighted in green. (page 4, line 169)

Point 4: Section 2.4.3: It is not clear from the paper on the basis of what considerations the parameters presented in Table 1 were chosen.

Response 4: All parameters were the same values as in the previous study. We have added the sentence below, which is highlighted in green in the text.

"All parameters in this section have the same values as in the previous study." (page 5, line 214)

Point 5: Section 4: Based on Figure 5, the authors compared methods using statistical methods but they are not described anywhere in the paper. It is also unclear what the asterisks in this figure mean. I assume that the asterisks indicate the level of statistical significance but the paper should describe it.

Response 5: Section 2.5, "Statistical Analysis," has been added to address this. The corrected parts are highlighted in green. (page 6, line 260)

Furthermore, Figure 5 (now Figure 4) has been corrected to include a description of the asterisks.

Reviewer 3 Report

The manuscript describes the results of using an RNN to detect saccade offset from EOG data. The authors state that that is currently mainly done using visual detection by experimenters.
Using neural networks for pattern recognition like this is a very reasonable and straightforward idea. However, I do think for it to be a publishable study significant work has to be done. Most importantly, the choice of this NN architecture needs to be justified or compared to other machine learning algorithms to detect the SO. Why did you use this RNN and how do you know it is the best choice? Many of the choices in the paper are not very well justified and either need further explanation or references. There are also some questionable statements and claims that you either need sources for or be more careful with your phrasing. I attached my further comments in the pdf.

Author Response

Responses to Reviewer 3 Comments

Point 1: Most importantly, the choice of this NN architecture needs to be justified or compared to other machine learning algorithms to detect the SO. Why did you use this RNN and how do you know it is the best choice? Many of the choices in the paper are not very well justified and either need further explanation or references. There are also some questionable statements and claims that you either need sources for or be more careful with your phrasing. I attached my further comments in the pdf.

Response 1: We have reworked the Introduction and included an explanation of why we used the RNN. The revised parts of the paper are highlighted in green. (pages 3, line 104)

Point 2: Section 1: Grammar mistake

Response 2: We have commissioned proofreading of the manuscript.

Point 3: Section 1

Response 3: Thank you very much for your constructive comments regarding Section 1. We have revised the Introduction in its entirety based on these comments.

Point 4: Section 2.4.1: Why? “500ms”

Response 4: We have included the below explanation, and highlighted the text in green.

"The width of the median filter was determined visually to minimize fluctuations in the value of the static state of the EOG data." (page 4, line 161)

Point 5: Section 2.4.1: Why? “The EOG data were downsampled to 250 Hz.”

Response 5: We have clearly stated that these values are the same as those in the PV method. The revised parts are highlighted in green. (page 4, line 169)

Point 6: Section 2.4.1: 0 ms long trial? “in a range of [0, 400 ms] (i.e., 101 time points)”

Response 6: The sentence has been corrected as follows and is highlighted in green in the text.

“The preprocessed EOG data were segmented into lengths of 400 ms (i.e., 101 time points), and one segmented data item was considered as a single trial.” (page 4, line 172)

Point 7: Section 2.4.1: 2500 earlier? “71809”

Response 7: We have moved the commented text to clarify the reduction in the number of trials. The revised parts are highlighted in green. (page 4, line 165)

Point 8: Section 2.4.1: Grammar “Since the input data, which were segmented EOG data (a range of [0, 400 ms]; 101 time points), T was 101.”

Response 8: We have revised this sentence, as follows, and have highlighted it in green.

“As the input data were single-trial EOG data, T was 101.” (page 4, line 193)

Point 9: Figure 1: Missing subscripts.

Response 9: We have corrected Figure 1.

Point 10: Figure 2: Needs to be improved.

Response 10: We have corrected Figure 2.

Point 11: Section 2.4.3: Why is the peak point saccade time? and why care about peak velocity? Why would you compare this to your algorithm? Needs further explanation and motivation.

Response 11: We have clearly stated the reason for using the PV method and have explained the methodology of the previous study. We have highlighted the revised parts in green. (page 5, line 212)

Point 12: Section 2.4.3: Then why do you do this?

Response 12: We have corrected the caption of Table 1 as shown below and highlighted it in green. “Summary of parameters of findpeaks() functions used in the PV method.” (page 6, line 244)

Point 13: Section 2.4.4: Why? “1 to 15 Hz”

Response 13: The values are the same as in the previous study [20].

Point 14: Section 3: What? Amount? “5.07 (SD = 24.0).”

Response 14: We have included units for all of the listed values and the corrected parts are highlighted in green. (page 6, line 274; page 7, line 285; page 7, line 286)

Point 15: Figure 5 amplitude looks the same in F4?

Response 15: The same values are used in Figures 4 and 5.

Point 16: Section 4: source? “Otherwise expressed, as the consistency of SO timing decreases, the mean amplitude decreases.”

Response 16: We have included an appropriate reference [10] and have highlighted the inclusion in green. (page 8, line 324)

Point 17: Section 4: Why did you chose this in the first place? “RNN-based”

Response 17: We have explained the reason for using an RNN in the fifth paragraph of the Introduction. The revised parts are highlighted in green. (page 3, line 104)

Point 18: Section 4: How? “in the future”

Response 18: We have specified future aims in the Discussion and Conclusions sections. The revised parts are highlighted in green. (page 9, line 355)

Round 2

Reviewer 1 Report

The authors have provided an improved version of their original manuscript. Since the revised manuscript has replied to my previous comments, I think the paper can be considered for publication.

Reviewer 2 Report

The authors have revised the paper well.  I think the paper can be accepted for publication.